# Activation of neural lineage networks and ARHGEF2 in enzalutamide-resistant and neuroendocrine prostate cancer and association with patient outcomes

Shu Ning[1], Jinge Zhao[1,6], Alan P. Lombard[1], Leandro S. D'Abronzo[1], Amy R. Leslie[1], Masuda Sharifi[1], Wei Lou[1], Chengfei Liu[1,2], Joy C. Yang[1], Christopher P. Evans[1,2], Eva Corey[3], Hong-Wu Chen[2,4], Aiming Yu[2,4], Paramita M. Ghosh[1,2,4,5] & Allen C. Gao[1,2,5✉]

## Abstract

**Background** Treatment-emergent neuroendocrine prostate cancer (NEPC) after androgen receptor (AR) targeted therapies is an aggressive variant of prostate cancer with an unfavorable prognosis. The underlying mechanisms for early neuroendocrine differentiation are poorly defined and diagnostic and prognostic biomarkers are needed.

**Methods** We performed transcriptomic analysis on the enzalutamide-resistant prostate cancer cell line C4-2B MDVR and NEPC patient databases to identify neural lineage signature (NLS) genes. Correlation of NLS genes with clinicopathologic features was determined. Cell viability was determined in C4-2B MDVR and H660 cells after knocking down *ARHGEF2* using siRNA. Organoid viability of patient-derived xenografts was measured after knocking down *ARHGEF2*.

**Results** We identify a 95-gene NLS representing the molecular landscape of neural precursor cell proliferation, embryonic stem cell pluripotency, and neural stem cell differentiation, which may indicate an early or intermediate stage of neuroendocrine differentiation. These NLS genes positively correlate with conventional neuroendocrine markers such as chromogranin and synaptophysin, and negatively correlate with AR and AR target genes in advanced prostate cancer. Differentially expressed NLS genes stratify small-cell NEPC from prostate adenocarcinoma, which are closely associated with clinicopathologic features such as Gleason Score and metastasis status. Higher *ARGHEF2*, *LHX2*, and *EPHB2* levels among the 95 NLS genes correlate with a shortened survival time in NEPC patients. Furthermore, downregulation of *ARHGEF2* gene expression suppresses cell viability and markers of neuroendocrine differentiation in enzalutamide-resistant and neuroendocrine cells.

**Conclusions** The 95 neural lineage gene signatures capture an early molecular shift toward neuroendocrine differentiation, which could stratify advanced prostate cancer patients to optimize clinical treatment and serve as a source of potential therapeutic targets in advanced prostate cancer.

## Plain language summary

Neuroendocrine prostate cancer is an aggressive subtype of prostate cancer that can arise in patients after treatment with drugs that suppress androgen hormone activity. Treatment options are limited for neuroendocrine prostate cancer and survival is poor, so it is important to identify markers to identify this subtype and how it relates to patient outcome. We use cells and patient datasets to identify a signature of 95 genes that can distinguish neuroendocrine prostate cancer from the more common prostate adenocarcinoma. Levels of a subset of these genes are associated with shortened survival time in patients. We show that one of these genes, ARHGEF2, helps to maintain survival and neuroendocrine features in prostate cancer cells. Our findings could help clinicians to identify patients with neuroendocrine prostate cancer and predict outcome in these patients, and help researchers to identify potential drug targets.

[1] Department of Urologic Surgery, University of California Davis, Sacramento, CA, USA. [2] UC Davis Comprehensive Cancer Center, University of California Davis, Sacramento, CA, USA. [3] Department of Urology, University of Washington, Seattle, WA, USA. [4] Department of Biochemistry and Molecular Medicine, University of California Davis, Sacramento, CA, USA. [5] VA Northern California Health Care System, Sacramento, CA, USA. [6] Present address: Department of Urology, West China Hospital, Sichuan University, Sichuan, China. ✉email: acgao@ucdavis.edu

Next-generation antiandrogen agents such as enzalutamide and abiraterone have been developed to suppress androgen biogenesis and block the activated AR signaling[1–4]. However, in recent years, it has become more prevalent for NEPC occurrence in CRPC patients who have received potent AR signaling inhibitors, which trigger the AR-independent tumor growth and neuroendocrine differentiation[5–7]. In neuroendocrine differentiation cancers such as small-cell lung cancer, various studies have demonstrated that transcriptional plasticity contributes to intratumoral heterogeneity and treatment resistance[8–10]. Prostate cancer cells upon the AR-targeted treatment acquire stem-like characteristics through tumor cell plasticity and molecular reprogramming, which contribute to transdifferentiation of neuroendocrine[11].

Neuroendocrine prostate cancer (NEPC) is a lethal phenotype of prostate cancer, characterized by immunobiological expression of NE markers including synaptophysin (SYP), chromogranin (CHGA), low expression of prostate-specific antigen (PSA), and androgen receptor (AR), and unresponsiveness to standard hormonal targeted treatments. Neuroendocrine differentiation gives rise to a more aggressive and malignant clinical phenotype, which has a higher incidence under the exposure to AR-directed therapies in prostate cancer patients[5,12–14]. Patients diagnosed with NEPC usually have dismal survival time and unfavorable prognosis[15,16]. The median survival for t-NEPC is approximately 20 months[14]. Given the rapid progression and malignancy of NEPC, there is an urgent need for early detection of neuroendocrine differentiation in CRPC patients for clinical diagnosis and therapeutic decision making.

Dissecting treatment-induced tumor heterogeneity and identifying biomarkers of neuroendocrine differentiation led to the successful translation to clinical diagnosis, prognosis, and therapies for prostate cancer patients. Recent studies have identified gene signatures regulating cell cycle progression as a prognostic predictor in NEPC[5,17]. In aggressive NEPC, a set of neuronal-related genes (BSN, CRMP1, GPRIN1, INA, MAST1, MYT1, RAB3C, SNAP25, UNC13A) were identified that confer to resistance to contemporary antiandrogen therapies[18]. Comparing AR-high metastatic CRPC, AR low, amphicrine and neuroendocrine PCa, 26-gene transcriptional signatures (including CELF3, PCSK1, SOX2, POU3F2, LMO3, NKX2-1, etc.) has been developed to distinguish variant phenotypes of metastatic castration-resistant prostate cancer (mCRPC)[19]. In different types of cancers, biomarker-targeted therapies have dramatically improved clinical outcomes. Therefore, the identification and validation of early emergent neural lineage markers that govern the response to antiandrogen resistances and neuroendocrine differentiation represents a fundamental unmet need.

In this study, we assessed the neural lineage pathway in enzalutamide-resistant C4-2B MDVR cell line and NEPC patients. We identified 95 gene neural lineage signatures (NLSs) underlying the molecular landscape of neural precursor cell proliferation, embryonic stem cell pluripotency, and neural stem cell differentiation/projection, which may represent a spectrum of neuroendocrine differentiation in advanced PCa. These NLS genes positively correlate with classical NE makers such as NSE, SYP, and CHGA and negatively correlate with AR and AR target genes in small-cell neuroendocrine group compared with prostate adenocarcinoma. Differentially expressed NLS genes can distinguish small-cell/NEPC from adenocarcinoma, indicating a diagnostic potential for CRPC patient stratification. Kaplan–Meier survival analyses revealed that the levels of ARHGEF2, LHX2, and EPHB2 expression among the 95 neural lineage genes were associated with shortened survival time. Further studies showed that the downregulation of ARHGEF2 gene expression suppresses cell viability and markers of neuroendocrine in enzalutamide-

resistant and neuroendocrine cells. Together, these findings suggest that treatment-emergent NEPC has a unique neural lineage network that can potentially serve as prognostic markers and can be targeted to treat enzalutamide-resistant and NEPC.

## Methods

**Cell culture**. C4-2B parental and enzalutamide-resistant (referred to as C4-2B MDVR) prostate cancer cells were obtained from the American Type Culture Collection (ATCC) and were cultured in RPMI 1640 supplemented with 10% fetal bovine serum (FBS), 100 units/ml penicillin and 0.1 mg/ml streptomycin. C4-2B MDVR cells were maintained in 20 µM enzalutamide for over 10 months. Enzalutamide purchased from Selleck Chemicals was dissolved in DMSO and stored at −20 °C. C4-2B parental cells were passaged alongside the resistant cells as the corresponding control. H660 cells were obtained from ATCC and were cultured in RPMI 1640 plus 5% FBS as the basic medium and were supplemented with 0.005 mg/ml insulin, 0.01 mg/ml transferrin, 30 nM sodium selenite, 10 nM hydrocortisone, 10 nM beta-estradiol, extra 2 mM L-glutamine. All cells were cultured in 37 °C humidified incubators with 5% $CO_2$.

**Cell transfection**. C4-2B MDVR and H660 cells were plated at 60–80% confluence in 60 mm dishes and transfected with 25 nM of small interfering RNA (siRNAs) targeting ARHGEF2 sequence (Catalog# 31215495), LHX2 sequence (Catalog# 312154951), EPHB2 sequence (Catalog# 312154948), or control siRNA (Catalog# 12935300) using Lipofectamine RNAiMAX transfection reagent (Invitrogen). The effects of siRNA knocking down were evaluated using qRT-PCR and western blot after 48 h transfection.

**RNA extraction and RNA sequencing**. Total RNA was isolated from C4-2B parental, MDVR cells by TriZOL reagent (Invitrogen). RNA integrity was assessed using RNA Nano 6000 Assay Kit of the Bioanalyzer 2100 system (Agilent Technologies). RNA sequencing libraries were generated using NEBNext Ultra RNA Library Prep Kit for Illumina (NEB, USA) according to the manufacturer's instruction and index codes were added to attribute sequences to each sample. cDNA fragments of preferentially 150–200 bp in length were selected using AMPure XP system (Beckman Coulter, USA). The index-coded samples were further clustered using PE Cluster Kit cBot-HS (Illumina) and sequenced on the Illumina platform. Paired-end clean reads were aligned to the reference human genome assembly (GRCh38/hg38) using the Spliced Transcripts Alignment to a Reference software. The gene expression level was calculated as FPKM (fragments per kilobase of transcript sequence per Millions base pairs sequenced). The hierarchical clustering analysis was used to cluster samples with similar expression patterns for different phenotypes of prostate cancer tumors.

**Real-time quantitative RT-PCR**. Total RNA was extracted from C4-2B parental, MDVR, H660 cells, LuCaP35CR, and LuCaP49 patient-derived xenograft (PDX) tumors using TriZOL reagent (Invitrogen Catalog# #15596018). cDNA was prepared using reverse transcriptase purchased from Promega. The cDNAs were subjected to quantitative PCR using SsoFast EvaGreen Supermix (Bio-Rad) according to the manufacturer's manual. Samples were run in triplicates on Bio-Rad CFX-96 real-time cycler. Each reaction was normalized by coamplification of GAPDH. Primers used for real-time PCR are as follows: ARHGEF2, 5′-TACCTG CGGCGAATTAAGAT-3′ (forward) and 5′-AAACAGCCCGAC CTTCTCTC-3′ (reverse); LHX2, 5′-CCGCACACTTCAACCAT GC-3′ (forward) and 5′-CCACGCCATTGTAGTAGGGAA-3′

(reverse); EPHB2, 5′-CGAGCCACGTTACATCA-3′ (forward) and 5′-TCAGTAACGCCGTTCACAGC-3′ (reverse); GAPDH, 5′-ATGAGTCCTTCCACGATACCA-3′ (forward) and 5′-GAAA TCCCATCACCATCTTCC-3′ (reverse).

**Cell lysate collection and western blotting**. Cells were harvested and lysed in RIPA buffer and the concentration was determined by Coomassie Plus Protein Assay (Thermo Scientific, Catlog#1856210). Protein extracts were resolved by SDS-PAGE, and proteins were transferred to nitrocellulose membranes. After blocking in 5% milk in PBS-0.1% Tween-20 for 1 h at room temperature, targeted proteins were detected by incubating the NC membrane with primary antibodies at 4 °C overnight. AR (441), 1:1000 dilution (Santa Cruz Biotechnology, Santa Cruz, CA); GEF-H1 (55B6) Rabbit mAb from Cell Signaling Technology (Catalog#4076); CHGA from Santa Cruz Biotechnology (Catalog#393941); SYP from Santa Cruz Biotechnology (Catalog#9116); GAPDH from Cell Signaling Technology (Catalog#2118). GAPDH was used as loading control. Following secondary antibody incubation, immunoreactive proteins were visualized by applying Immobilon Crescendo Western HRP substrate (Millipore, Catalog#WBLUR0500).

**Gene Set Enrichment Analysis**. Gene expression results were subjected to Gene Set Enrichment Analysis (GSEA) to determine patterns of pathway activity in different group of samples. GSEA using GSEA software from the Broad Institute was used to identify biological function pathways based on Molecular Signature Database. The pathway analysis was performed using pathway gene sets annotated by Gene Ontology (GO), Kyoto Encyclopedia of Genes and Genomes, and PathCards (Pathway Unification Database) (https://pathcards.genecards.org/). Pathways enriched with normalized enrichment score (NES) and a nominal $p$ value lower than 0.05 and FDR $q$ value lower than 0.25 were considered to be significant.

**PDX tumor xenografts and organoid culture**. All animals used in this study received humane care in compliance with applicable ethnic regulations, policies, and guidelines relating to animals. Animal experiments were conducted in accordance with animal protocols approved by the Institutional Animal Care and Use Committee of the University of California Davis. Male C.B17/lcrHsd-Prkdc-SCID mice from ENVIGO were used for the PDX xenograft at 4–6 weeks of age and housed in standard cages in ultraclean barrier facilities. The housing conditions for the mice were as follows: 12-h light-dark cycle; temperature was 68–79 °F; and humidity was 30–70%. Animals demonstrating sickness or severe stress were euthanized and excluded. Otherwise, all data were included in the study.

LuCaP35 CR and LuCaP49 PDX models were obtained from the University of Washington and established at the UC Davis. Four- to six-week-old male C.B17/lcrHsd-Prkdc-SCID mice (ENVIGO) were surgically castrated and implanted subcutaneously into the flanks with approximately 2 mm³ LuCaP35 CR tumor pieces after 1 week. LuCaP49 tumor pieces were directly implanted subcutaneously to 4–6-week-old male C.B17/lcrHsd-Prkdc-SCID mice (ENVIGO) and tumors were collected after 5 weeks of growth. LuCaP35CR and LuCaP49 PDX tumor specimens were collected from the corresponding donor mice and subjected to RT-qPCR and organoid culture.

LuCaP PDX-derived tumor tissues were collected and washed twice in cold PBS, and subjected to dissection to 2–4 mm³ with scalpel blade. Tumors were digested using collagenase IV (STEMCELL) in a 60-mm² petri dish and incubated in 37 °C humidified incubators with 5% $CO_2$ for 15–30 min until tumor

cells were dispersed in the digestive medium. Advanced DMEM medium supplemented with 1X GlutaMAX (Gibco), 1 M HEPES (Gibco) and 100 u/ml penicillin and 0.1 mg/ml streptomycin were added to the cell suspension, and then filtered through 40-μm cell strainers to obtain single cell suspension. The cells were centrifuged at 500 g for 3 min and the pellet was resuspended in an advanced DMEM complete medium containing GlutaMAX (Gibco), 100 units/ml penicillin, 0.1 mg/ml streptomycin, B27 (Gibco), N-Acetylcysteine (Thermo Scientific), Human Recombinant EGF (Thermo Scientific), Recombinant FGF-10 (Invitrogen), A-83-01 (Tocris), SB202190 (Bioscience), Nicotinamide (Thermo Scientific), dihydrotestosterone (Sigma), PGE2 (Bioscience), Noggin (Thermo Scientific) and R-spondin (R&D Systems)[20,21]. Tumor cell pellet was seeded in 96-well plate with Matrigel diluted with 1:3 ratio of ADMEM complete medium and incubated in 37 °C humidified incubators with 5% $CO_2$ for 15 min to solidify the 3D Matrigel complex. Then ADMEM complete medium mixed with corresponding treatment was added to each well. Viability of the organoids was analyzed using CellTiter Glo Luminescent assay (Promega) and visualized by immunofluorescence using LIVE/DEAD® Viability/Cytotoxicity Assay Kit (Thermo Scientific) according to the manufacturer's protocol.

**Statistics and reproducibility**. A two-sided Wilcoxon test was used to analyze the significance of gene expression between two groups. A $p$ value less than 0.05 was considered significant ($*p < 0.05$, $**p < 0.01$, $***p < 0.001$) unless otherwise indicated. Kaplan–Meier curves were plotted to analyze the survival probability of the two groups. A log-rank test was used to compare the overall survival or disease-free survival between the two groups of different gene expression level. A $p$ value lower than 0.05 indicated that the two groups differed significantly in overall survival or disease-free survival.

**Reporting summary**. Further information on research design is available in the Nature Research Reporting Summary linked to this article.

## Results

**Neural lineage network enriched in enzalutamide-resistant prostate cancer cells**. It has been well established that the alteration of the neural-associated molecular landscape in castration-resistant tumors may contribute to the androgen and antiandrogens indifferences and the neuroendocrine progression after the AR-targeted treatment[5,7,13]. However, the overall neural lineage network underlying the end-stage emergences of these neuroendocrine markers remains unknown. We have previously generated an enzalutamide-resistant cell subline named C4-2B MDVR from C4-2B cell line through a long-term culture of C4-2B cells in the presence of increasing doses of enzalutamide[22]. Recent studies have shown multiple characteristics presented in C4-2B MDVR cells including overexpression of AR/AR-V7, Wnt signaling activation such as Wnt5a and WLS, and increased expression of markers of neuroendocrine such as NSE and CHGA[23,24], suggesting that C4-2B MDVR cells may represent multiple cellular lineages including neuroendocrine.

To dissect the molecular changes associated with neuroendocrine differentiation in C4-2B MDVR cells, we first analyzed several well-known NED gene signatures in the transcriptome of C4-2B MDVR cells. Genes encoding several of the classic NE markers such as ENO2, CHGA, and SYP were upregulated in C4-2B MDVR cells compared to the parental C4-2B cells (Fig. 1a). Since neuroendocrine cells may derive from neural crest cells and embryonic stem cells, we analyzed the transcriptome of C4-2B MDVR cells for the pathways relative to neural stem cell

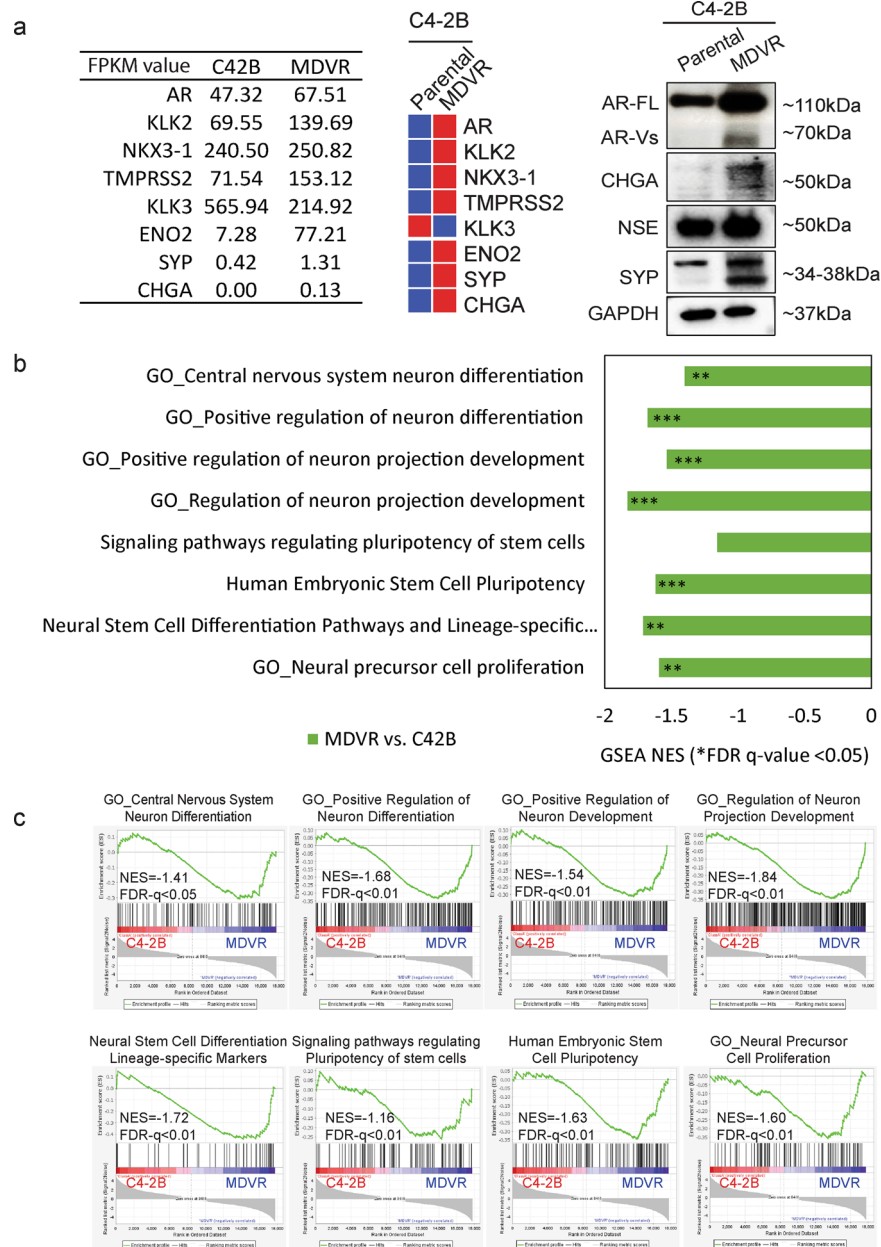

**Fig. 1 Neural lineage pathways enriched in enzalutamide-resistant prostate cancer C4-2B MDVR cells. a** Gene and protein expression of the classical NE markers in C4-2B MDVR cells compared to the parental C4-2B cells. **b** Summarized GSEA analysis of enriched pathway or gene sets in C4-2B MDVR cells compared to the parental cells. The altered gene sets from Gene Ontology and PathCards were generated by GSEA software. **c** Enrichment plots of GSEA analyses for the neural lineage pathways in C4-2B MDVR cells compared with C4-2B parental cells. NES, showing normalized enrichment score (NES) and corresponding significance for the eight pathways. *FDR *q* value <0.05, **FDR *q* value <0.01, ***FDR *q* value <0.001. NE neuroendocrine, GSEA Gene Set Enrichment Analysis, FDR false discovery rate.

proliferation and neuron differentiation and projection pathways (Supplementary Table 1) in order to determine if neural lineage genes are enriched in C4-2B MDVR cells. Among these pathways, we found that eight pathways are significantly enriched (NES < −1.4, FDR *p* value <0.05) in the transcriptome of C4-2B MDVR cells compared to that of parental C4-2B cells (Fig. 1b, c). The eight enriched pathways fall into four major categories, including neural stem cell differentiation, neural precursor cell proliferation, embryonic stem cell pluripotency, and neuron differentiation/projection (Fig. 1b, c). Our data suggest that neural lineages may emerge from enzalutamide-resistant cells, which may adopt a cell plasticity of early or intermediate stage of neuroendocrine differentiation.

**Neural lineage pathways and gene signatures in clinical neuroendocrine prostate cancer.** To examine if the neural lineage pathways identified in C4-2B MDVR cells are presented in patients with NEPC, we performed GSEA pathway enrichment analyses on small-cell and NE prostate cancer cohorts from two multi-institutional prospective studies, mCRPC[13], and treatment-emergent small-cell neuroendocrine prostate cancer (t-SCNC)[5]. The analyses were based on a total of 49 mCRPC cases, including 34 CRPC and 15 NEPC samples in Beltran's study, and a total of 119 samples including 15 pure small cells, 6 mixed small cells, and 98 adenocarcinoma samples from Aggarwal's study. We found that the eight neural lineage pathways enriched in C4-2B MDVR cells were also enriched in these NEPC patient datasets,

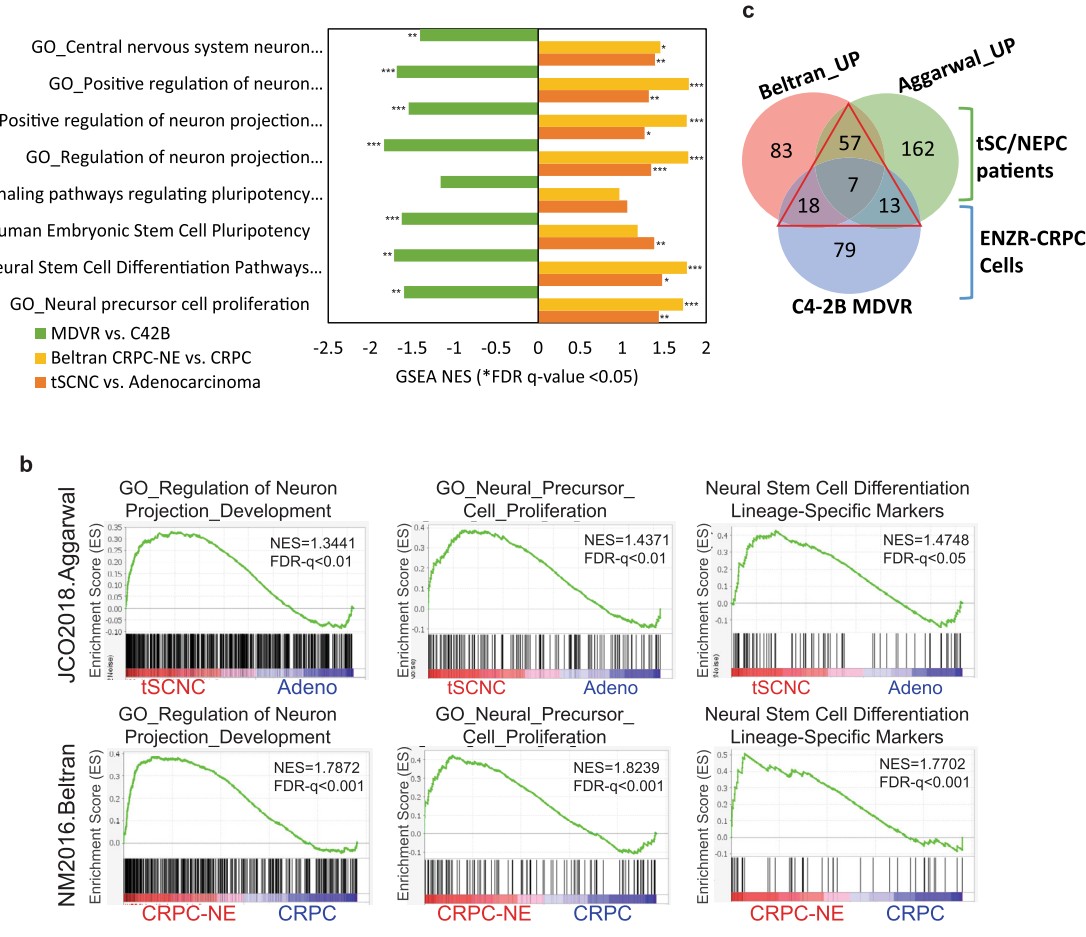

**Fig. 2 Neural lineage pathways both enriched in clinical neuroendocrine prostate cancer patients and enzalutamide-resistant prostate cancer cell model. a** Parallel comparison of neural lineage network in clinical NEPC patients and enzalutamide refractory prostate cancer cell model C4-2B MDVR. **b** GSEA enrichment plots of the neural lineage gene sets between the indicated groups showing NES scores and corresponding significance in the C4-2B MDVR vs. C4-2B parental cells (green), NEPC patients vs. CRPC patients (yellow), and t-SCNC patients vs. prostate adenocarcinoma (orange). *FDR *q* value <0.05, **FDR *q* value <0.01, ***FDR *q* value <0.001. **c** Venn diagram showing the number of upregulated genes from neural lineage pathways among Beltran, Aggarwal's patient datasets (Wilcox rank-sum test *p* < 0.05), and C4-2B MDVR cells (|log 2-fold change|>1, FPKM value >1). NEPC neuroendocrine prostate cancer, GSEA Gene Set Enrichment Analysis, t-SCNC treatment-emergent small-cell neuroendocrine prostate cancer.

including Neural Stem Cell Differentiation Pathways and Lineage-Specific Markers, Neural Precursor Cell Proliferation, Central Nervous System Neuron Differentiation, and Neuron Projection Development from Gene Ontology (Fig. 2a, b and Supplementary Fig. 1). The results from patient datasets suggest that the neural lineage pathways identified in C4-2B MDVR cells are also enriched in NEPC patients.

Having demonstrated that neural lineage programs are enriched in NEPC, we next analyzed the signature panel of neural lineage genes that are upregulated in the neural lineage programs. Based on the enriched neural lineage pathways aforementioned, we performed the Wilcox rank-sum test analyses and found that 239 genes were differentially upregulated in t-SCNC groups in the Aggarwal study and 165 genes in CRPC-NE cohorts in the Beltran study (*p* < 0.05). In parallel, we found 1060 genes upregulated in C4-2B MDVR cells (|log 2-fold change| >1, FPKM value >1). Venn diagram (Fig. 2c, displayed the commonly upregulated genes from the two patient datasets and C4-2B MDVR cells, and a collective 95 unique genes were identified as the gene panel of NLS (Fig. 2c and Supplementary Data 1). The neural lineage 95-gene panel was comprised of commonly upregulated genes in both Aggarwal and Beltran study (57 genes) and shared 7 genes and upregulated genes in C4-2B MDVR overlapped with either of the patient cohorts (13 and 18

genes for overlapped in MDVR vs. Aggarwal and MDVR vs. Beltran). Unsupervised hierarchical cluster analysis of 95 differentially expressed NLS genes (95 NLS genes) in Beltran CRPC-NE and Aggarwal t-SCNC patients were presented in Fig. 3a, b, which markedly clustered in the small-cell neuroendocrine groups. A correlation plot revealed that 95 neural lineage genes were positively correlated with these 29 NE markers and inversely correlated with AR and classical AR-targeted genes in both Beltran CRPC-NE and Aggarwal t-SCNC patients (Supplementary Fig. 2a, b). GSEA analysis also showed a significant enrichment of 95-gene NLSs in NEPC groups of these two patient datasets (Fig. 3c, d), which is consistent with the enrichment of 29 genes from the defined NEPC classifier[13]. Collectively, our analyses suggested that the 95 NLS genes (95 NLS) provide a molecular background giving rise to neuroendocrine differentiation in enzalutamide-resistant prostate cancer.

**The 95 neural lineage gene signatures stratified NEPC from CRPC.** We next determined if the 95 neural lineage gene signatures (NLS) could be used to stratify NEPC from CRPC in two advanced CRPC databases[19,25]. A hierarchical clustering heatmap demonstrated that NLS genes significantly clustered in the group of small cell or adenocarcinoma with NE features in the

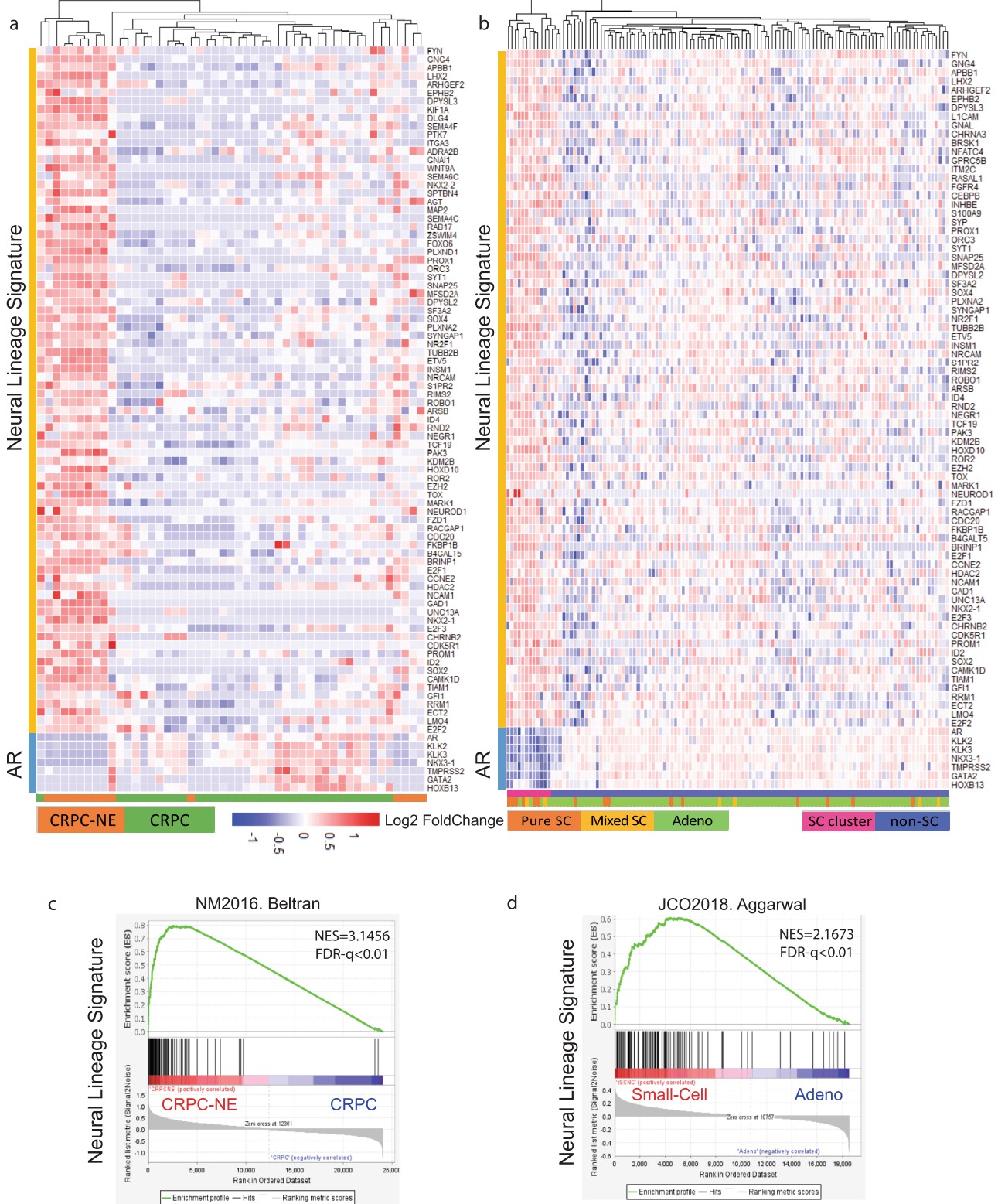

**Fig. 3 Identified neural lineage signature genes upregulated in CRPC-NE patients. a** Unsupervised hierarchical cluster of 95 differentially genes expressed in Beltran CRPC-NE cohorts. Pathology classification (CRPC vs. CRPC-NE) is indicated in the annotated track next to the heatmap. **b** GSEA enrichment plots with NES scores and FDR *q* values shown. *FDR *q* value <0.05, **FDR *q* value <0.01, ***FDR *q* value <0.001. **c** Hierarchical cluster analysis of 95 differentially expressed neural lineage genes in Aggarwal t-SCNC cohorts. Pathology classification (Pure/Mixed small cell/Adenocarcinoma) and molecular cluster of small-cell and non-small-cell groups is indicated in the annotation track next to the heatmap. **d** Enrichment plots for neural lineage signature in Aggarwal t-SCNC cohorts with NES scores and FDR *q* values shown. *FDR *q* value <0.05, **FDR *q* value <0.01, ***FDR *q* value <0.001. CRPC castration-resistant prostate cancer, CRPC-NE castration-resistant prostate cancer with neuroendocrine features, GSEA Gene Set Enrichment Analysis, NES normalized enrichment score.

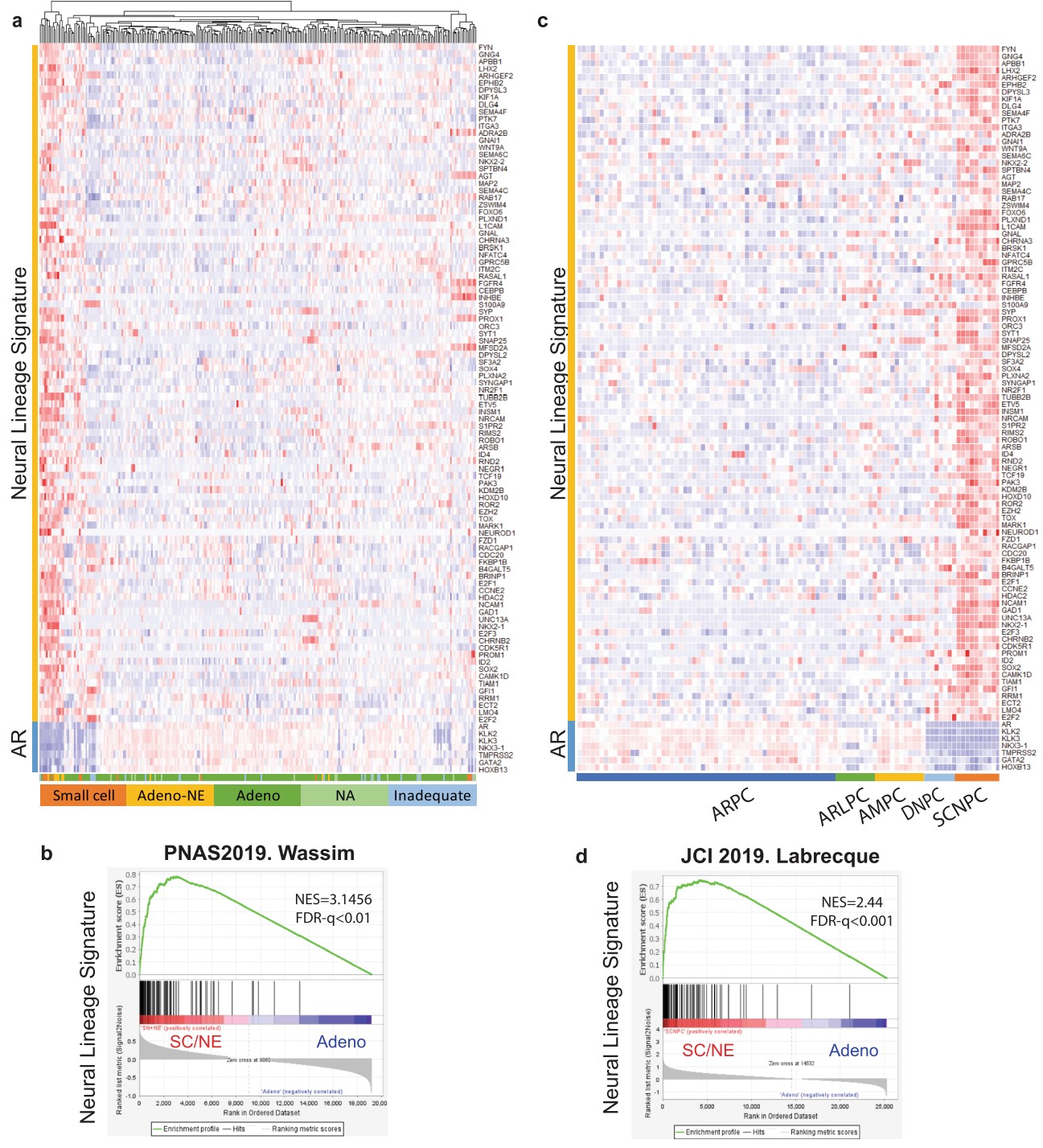

**Fig. 4 The 95 NLS gene signatures stratify prostate cancer with neuroendocrine differentiation from prostate adenocarcinoma. a** Hierarchical cluster of neural lineage signature genes in Wassim small-cell cohorts. Pathology classification (small cell, adenocarcinoma with neuroendocrine (NE) features, adenocarcinoma, not available and inadequate for the diagnosis) are indicated in the annotation track next to the heatmap. **b** GSEA enrichment plots for neural lineage signature in small cell and adenocarcinoma with NE feature groups with NES and FDR $q$ values shown. **c** Hierarchical cluster heatmap of neural lineage signature genes in Labrecque cohorts. Molecular characteristic classification (AR high (ARPC), AR low (ARLPC), amphicrine expression of both AR and NE markers (AMPC), double-negative expression of both AR and NE markers (DNPC), and small-cell neuroendocrine SCNPC) are indicated in the annotation track next to the heatmap. **d** GSEA enrichment plots of neural lineage signature genes in Labrecque cohorts with NES score and FDR $q$ values. NLS neural lineage signature, GSEA Gene Set Enrichment Analysis. *$p < 0.05$, **$p < 0.01$, ***$p < 0.001$.

Abida-Wassim cohort (Fig. 4a, b), which align with upregulation patterns of the aforementioned Beltran NEPC classifier genes and downregulation of AR related genes (Supplementary Fig. 3a). We also analyzed the NLS in the Labrecque study including refractory metastatic CRPC specimens (AR-positive tumors (ARPC, $n = 59$),

AR-low tumors (ARLPC, $n = 9$), amphicrine tumors expressing both AR and NE markers (AMPC, $n = 11$), double-negative tumors (DNPC, $n = 7$), and tumors with small cell or NE makers without AR features (SCNPC, $n = 10$))[19]. The heatmap of NLS genes demonstrated a clear distinction between the subtypes of

mCRPC specimens with GSEA enrichment significance (NES = 2.44, FDR $q$ value <0.01) (Fig. 4c, d), which was also consistent with the expression pattern of classic NED markers and AR target genes (Supplementary Fig. 3b). We also examined our NLS genes in an experimental model, TLT331R NEPC tumor established after castration and relapsed after 24–32 weeks[26]. As shown in Supplementary Fig. 4, top upregulated NLS genes were displayed in the unsupervised hierarchical clustering heatmap, which aligned well with NE markers and negatively correlated AR target genes. GSEA enrichment analyses also showed that NLS genes were significantly enriched in TLT331R NEPC tumor groups compared with prostate adenocarcinoma groups. In summary, these data indicated that these differentially expressed NLS genes could stratify prostate cancer with neuroendocrine differentiation from prostate adenocarcinoma.

**Higher levels of ARHGEF2, EPHB2, and LHX2 expression correlate with poor survival in castration-resistant prostate cancer.** Among the genes from the 95 NLS, we further analyzed 7 of the neural lineage genes (*ARHGEF2, EPHB2, LXH2, DPYSL3, EPHB2, FYN,* and *GNG4*) shared among C4-2B MDVR cells, Beltran, Aggarwal, Abida-Wassim and Labrecque datasets[5,13,19,25]. Our data showed that all the seven genes were upregulated in CRPC-NE/small-cell groups compared to the CRPC-adeno group across the four databases (Fig. 5a). To determine whether the seven NLS genes were associated with survival in prostate cancer patients, we further conducted the Kaplan–Meier survival analysis and log-rank test to determine the correlation of expression of the neural lineage genes to the overall survival in prostate cancer patients. In 75 out of 266 prostate cancer patients with overall survival of the first-line AR-targeted inhibitors treatment in the Abida-Wassim study[25], higher expression of *ARHGEF2* ($p = 0.041$), *LHX2* ($p = 0.0091$), and *EPHB2* ($p = 0.15$) showed correlation with shortened overall survival time (Fig. 5b), while *DPYSL3, EPHB2, FYN,* and *GNG4* did not positively correlate with shortened overall survival. We also performed the Kaplan–Meier survival analysis on 148 patients with both disease-free survival information and RNA sequencing data available from the MSKCC study, which included 131 primary tumors and 9 metastases[27]. The data revealed that higher expression of *ARHGEF2* ($p = 0.011$), *LHX2* ($p = 0.0091$), and *EPHB2* ($p = 0.0019$) correlate with shorter disease-free time, respectively (Fig. 5c), while *DPYSL3, EPHB2, FYN,* and *GNG4* did not reach statistical significance (data not shown). Collectively, these results suggest the potential of *ARHGEF2, LHX2,* and *EPHB2* as indicators of poor survival for advanced prostate cancer.

**Downregulation of ARHGF2 expression suppresses viability and neuroendocrine markers of C4-2B MDVR and H660 cells.** We analyzed *ARHGEF2, LHX2,* and *EPHB2* gene expression in the RNA sequencing data from GSE154576 (DeLucia et al., 2021). As shown in Fig. 6a, gene expression of *ARHGEF2, LHX2,* and *EPHB2* were significantly upregulated in NEPC MSKCC-EF1 and H660 compared to 22RV1 and LNCaP95 cell lines. Quantification of *ARHGEF2, LHX2,* and *EPHB2* mRNA levels verified that these three genes increased in enzalutamide-resistant C4-2B MDVR compared with C4-2B parental cells, and further increased in H660 cells (Fig. 6b). We next focused on the effect of *ARHGEF2* on cell growth and neuroendocrine differentiation by knocking down *ARHGEF2* expression using siRNA in C4-2B MDVR and H660 cells. Knocking down of *ARHGEF2* expression downregulates CHGA, NSE, and SYP protein expression (Fig. 6c) and inhibits cell viability (Fig. 6d) in both C4-2B MDVR and H660 cells. Furthermore, we analyzed LuCaP49 neuroendocrine PDX tumors and LuCaP35CR castration-resistant PDX tumors for

their expression of AR-targeted genes and NED markers and found that LuCaP49 tumors express higher levels of NE markers such as CHGA, NSE, and SYP than LuCaP35CR tumors (Fig. 6e), consistent with the characteristics of NED for LuCaP49 and CRPC for LuCaP35CR[28]. We also found that ARHGEF2 mRNA levels were much higher in LuCaP49 tumors than LuCaP35CR tumors (Fig. 6e). Knocking down ARHGEF2 expression through siRNA significantly inhibits the viability and growth of organoids derived from LuCaP49 PDX tumors (Fig. 6f). Collectively, these data suggest that *ARHGEF2* could serve as a potential therapeutic target for NEPC.

**Discussion**
In the present study, our comprehensive transcriptomic analyses of neural lineage pathways in enzalutamide-resistant CRPC cell line and NEPC patient database identified a 95-gene transcript panel as NLS. The expression of these NLS genes was positively correlated with classic NE markers such as CHGA, SYP, and NES, and was negatively correlated with PSA, AR, and AR target genes. Across different patient databases, these 95 NLS genes can distinguish prostate cancer of neuroendocrine differentiation from prostate adenocarcinoma. Among the 95 NLS genes, the expression of *ARHGEF2, LHX2,* and *EPHB2* is close associated with clinicopathologic features and shortened disease-survival outcomes in prostate cancer patients. Further studies showed that the downregulation of *ARHGEF2* gene expression suppresses cell viability and markers of neuroendocrine in enzalutamide-resistant and neuroendocrine cells. Collectively, these findings suggest that treatment-emergent NEPC has a unique neural lineage network that can potentially serve as prognostic markers and be used as a target to treat enzalutamide-resistant and NEPC.

Cumulative clinical evidence shows that next-generation AR-targeted treatment induces the emergence of neuroendocrine differentiation with small-cell histology and positive neuroendocrine markers[5,16,29–31]. Experimental studies also have shown that prostate adenocarcinoma cell line LNCaP can acquire neuroendocrine features under the long-term treatment of enzalutamide[5,13,23,24,32]. In alignment with these studies, we found that the eight neural lineage pathways including neural stem cell differentiation pathways and lineage-specific markers, neural precursor cell proliferation, central nervous system, neuron differentiation, and neuron projection development are significantly enriched in both enzalutamide-resistant C4-2B MDVR cells and in CRPC patients with NE features. These eight enriched pathways fall into four major categories, including neural stem cell differentiation, neural precursor cell proliferation, embryonic stem cell pluripotency, and neuron differentiation/projection, suggesting that neural lineages may emerge from enzalutamide-resistant cells, which may adopt cell plasticity of early or intermediate stage of neuroendocrine differentiation.

Several of the gene signatures representing NE features have been reported and used to guide a more precise molecular characterization of neuroendocrine differentiation in prostate cancer. Beltran et al. reported a 70-gene NE classifier based on genomic, transcriptomic, or epigenomic status, which has been shown clustering in CRPC-NE groups across 636 samples of four different patients' data (Beltran2016[13], SU2C/PCF[33], WCMC[34,35], and TCGA[36]). The integrated NE score has also been examined using different methods, 49 genes by ROC-curve selection, 67 genes with AR signaling genes excluded, and 12 genes by After Feature Selection strategy[13]. Although not completely the same across the different methods, these gene sets can well distinguish NE patients versus prostate adenocarcinoma. In t-SCNC dataset, Aggarwal et al. identified 61 genes differentially expressed in t-SCNC clustered samples, among which 25 transcriptional factors

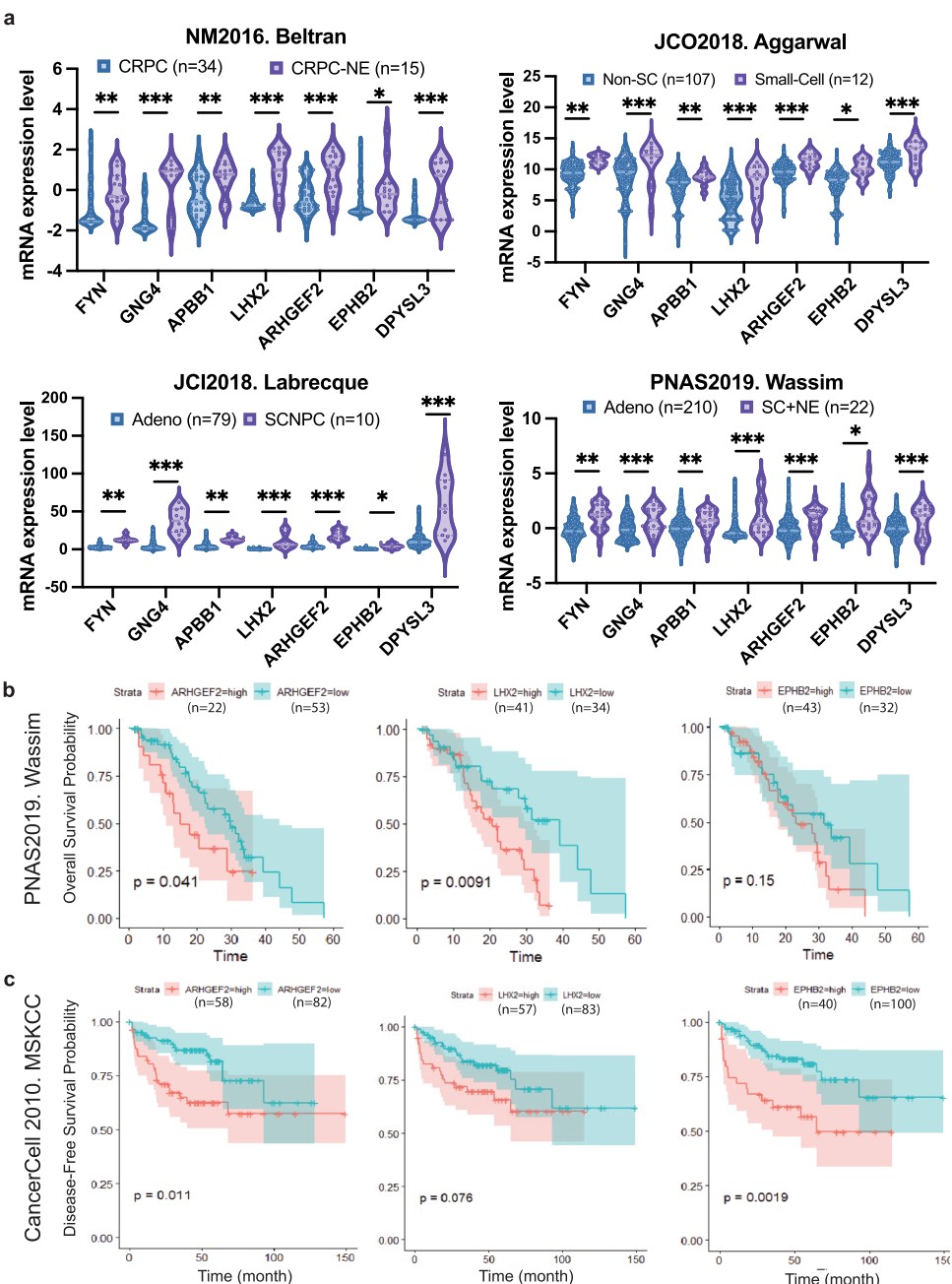

**Fig. 5 Upregulated neural lineage genes correlate with poor overall survival in prostate cancer patients. a** Gene expressions of the seven neural lineage signature genes were presented in the indicated groups of Beltran, Aggarwal, Labrecque, and Wassim cohorts. Gene expression from CRPC patients in Beltran study, prostate adenocarcinoma samples from Aggarwal, Labrecque, and Wassim study were presented in blue, while that of CRPC-NE samples from Beltran study, small-cell neuroendocrine prostate cancer from Aggarwal, Labrecque and Wassim studies was presented in purple. CRPC castration-resistant prostate cancer; CRPC-NE castration-resistant prostate cancer with neuroendocrine features; Adeno; non-SC non-small-cell prostate adenocarcinoma; Small-Cell small-cell neuroendocrine prostate cancer; prostate adenocarcinoma; SCNPC small-cell neuroendocrine prostate cancer; SC + NE small-cell neuroendocrine prostate cancer. *$p < 0.05$, **$p < 0.01$, ***$p < 0.001$, using unpaired *t*-test (median, quartiles and distribution of all data points presented). **b** Kaplan–Meier analysis of neural lineage signature genes including *ARHGEF2*, *LHX2*, and *EPHB2* for the overall survival in Wassim cohorts. **c** Kaplan–Meier analysis of neural lineage signature genes for disease-free survival in MSKCC cohorts using log-rank test.

were top expressed by MARINa-inferred algorithm[5]. Harrison, et al. reported a meta-signature to describe molecular features of prototypical primary small-cell prostate cancer[18], which may differ from the aforementioned treatment-induced NE differentiation in advanced CPRC patients. Similarly, 212 genes have been used to profile SC/NE patients compared with treatment-naïve primary prostate cancer[37]. More efforts are spared to characterize SC/NE features for de novo NEPC, metastatic CRPC,

or treatment-induced CRPC. In our study, 95 neural lineage genes were identified based on the biological function that regulates precursor cell proliferation, central nervous system neuron differentiation, and neuron projection development, providing additional molecular background and giving rise to neuroendocrine differentiation in enzalutamide-resistant prostate cancer. Besides, NLS genes were highly correlated with NE signature genes identified by Beltran and Aggarwal, which suggests that the 95

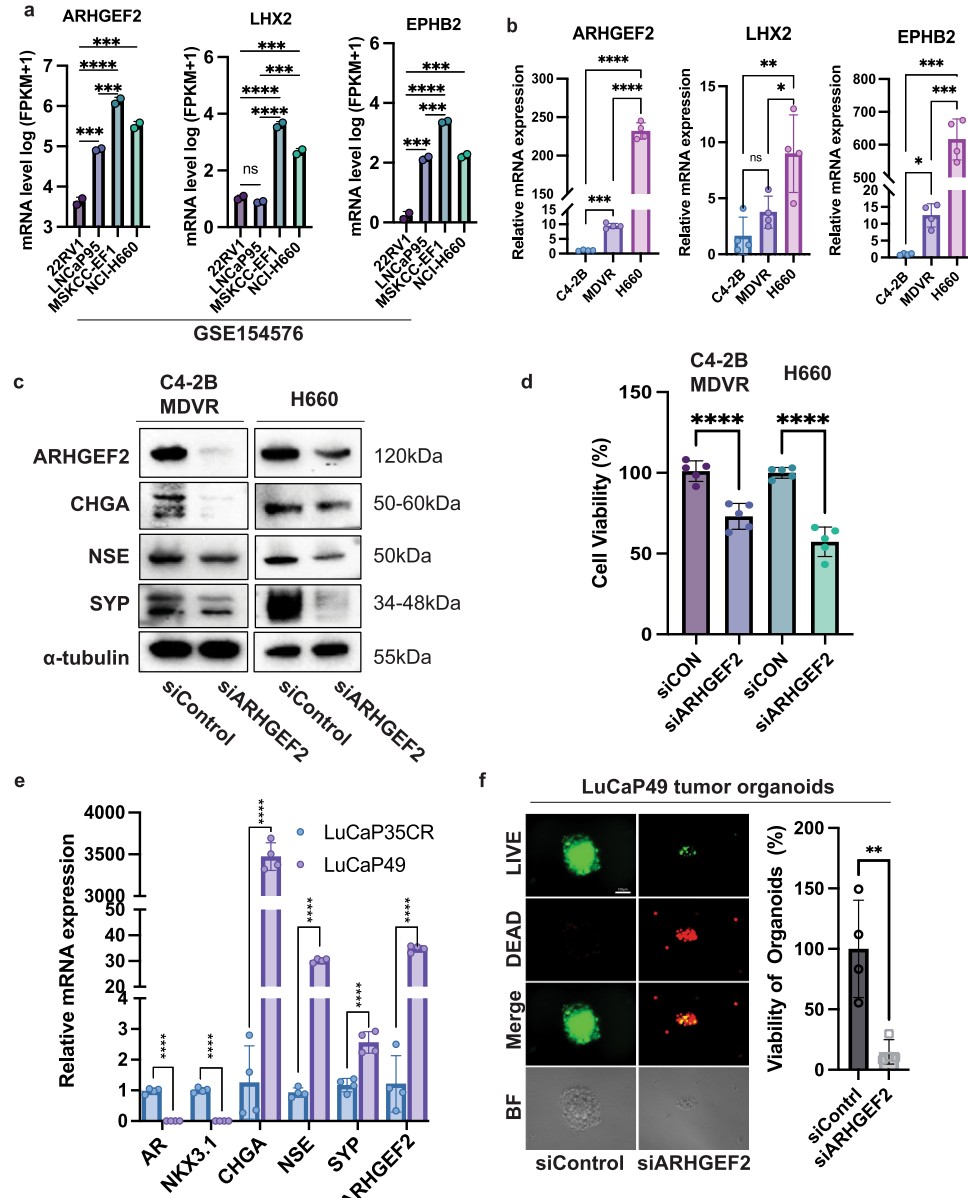

**Fig. 6 Downregulating neural lineage marker *ARHGEF2* gene expression decreased C4-2B MDVR and H660 cell viability. a** FPKM value of *ARHGEF2*, *LHX2* and *EPHB2* gene expression in 22RV1, LNCaP95, MSKCC-EF1 and H660 cells from GSE154576 dataset (*n* = 2). **b** Relative mRNA level of *ARHGEF2*, *LHX2*, and *EPHB2* in C4-2B parental, MDVR, and H660 cells (*n* = 4). **c** Knocking down *ARHGEF2* by siRNA decreased the expression of the markers of neuroendocrine including CHGA, NSE, and SYP in C4-2B MDVR and H660 cells. C4-2B MDVR and H660 cells were transfected with siRNAs for *ARHGEF2* for 48 h, total cell lysates were collected and subjected to western blot analysis. **d** Knocking down *ARHGEF2* by siRNA inhibited cell viability of C4-2B MDVR and H660 cells. C4-2B MDVR and H660 cells were transiently transfected with siRNAs targeting *ARHGEF2* and viable cells were quantified using CellTiter Glo assay (*n* = 5). **e** qPCR analysis of AR targets and neuroendocrine markers and *ARHGEF2* in NEPC LuCaP49 patient-derived xenograft (PDX) tumors and prostate adenocarcinoma LuCaP35CR tumors (*n* = 3). **f** Downregulation of *ARHGEF2* by siRNA inhibits the growth of organoids derived from LuCaP49 PDX tumors. LuCaP49 organoids were seeded in a 96-well plate in the format of 3D Matrigel and then transfected with 50 nM *ARHGEF2* siRNA and cultured for 14 days. The viability of the organoids was analyzed by CellTiter Glo and visualized by LIVE/DEAD staining (*n* = 4). Green = Calcein staining of live cells, Red = Ethidium homodimer-1 staining of dead cells. Scale bar 100 μm. *$p < 0.05$, **$p < 0.01$, ***$p < 0.001$ (mean and SD in **a**, **b**, **d**, **e**, **f**) using one-way ANOVA with multiple-comparisons test.

NLS genes can be used to stratify NEPC from CRPC in advanced prostate cancer.

Apart from NE signature genes identified by different patient samples and methods, more studies are focusing on master regulators for neuroendocrine differentiation and developing potential therapeutics targeting these genes. *MYCN* has been found to be an oncogenic gene with amplification and overexpression in NEPC, which concurrent with *AURKA* gene amplification, contributes to the treatment-emergent NEPC after

hormonal treatment[38–40]. Based on the interaction of *AURKA* with *MYNC*, *AURKA* inhibitors such as alisertib have been developed in clinical studies to target *AURKA-MYCN* axis and treat NEPC[34,41]. Furthermore, *MYCN* interacts with *EZH2* and downregulates AR signaling in the NEPC molecular program, and *EZH2* inhibitors provide the potential to treat NEPC patients[42]. BRN2 (encoded by *POU3F2*) drives neuroendocrine differentiation in treatment-resistant CRPC via SOX2 and has been also reported as a potential therapeutic target to treat

NEPC[43]. In our study, we demonstrated *ARHGEF2*, *LHX2*, and *EPHB2*, highly expressed in patients with NE features. GEF-H1 (encoded by *ARHGEF2*) is a Rho-specific guanine nucleotide exchange factor that determines the neuron genesis in the embryonic murine cortex[44]. *ARHGEF2* promotes cell migration and invasion in gastric and prostate cancer via RhoA/ROCK activation[45,46]. *LHX2* regulating progenitor differentiation and proliferation[47,48] was also found to be overexpressed in small-cell lung cancer and prostate cancer[49,50]. *EPHB2*, a receptor tyrosine kinase, promotes tumor progression in glioblastoma, colorectal cancer, and hepatocellular carcinoma[51–53]. To test the potential of NLS genes as therapeutic targets, *ARHGEF2* knocking down by siRNAs inhibited the biomarkers of NEPC such as CHGA, NSE, and SYP, as well as cell viability and growth of tumor organoids derived from NEPC tumors, suggesting that targeting the critical NLS genes could serve as potential therapeutic strategies for the treatment of NEPC.

In summary, our study focused on the common transcriptomic changes of neuroendocrine differentiation through the comprehensive analyses of four NEPC patient datasets and enzalutamide treatment CRPC cell model and identified the 95 NLS genes that can distinguish prostate cancer of neuroendocrine differentiation from prostate adenocarcinoma. Among the 95 NLS genes, the expression of *ARHGEF2*, *LHX2*, and *EPHB2* is close associated with shortened disease survival in prostate cancer patients. Further studies showed that the downregulation of *ARHGEF2* gene expression suppresses cell viability and markers of neuroendocrine in enzalutamide-resistant and neuroendocrine cells. These NLS genes could be utilized in future studies as biomarkers to better stratify patients based on CRPC subtypes and possibly guide the decision making for patients, and also to develop novel therapeutic strategies for the management of NEPC.

## Data availability

Tumor sample information and corresponding clinical characteristics from Beltran cohort castration-resistant neuroendocrine prostate cancer[13], Stand Up 2 Cancer/Prostate Cancer Foundation-funded West Coast Prostate Cancer Dream Team[54], and Abida-Wassim cohort metastatic castration-resistant prostate cancer[25] were downloaded from cBioPortal for Cancer Genomics (www.cbioportal.org). Whole-genome RNA sequencing for treatment-resistant metastatic castration prostate cancer[19] was obtained from NCBI's Gene Expression Omnibus (GEO) using the accession number GSE126078. Gene expression and clinical information from The Cancer Genome Atlas Research Network[36] were downloaded from the cBioPortal for Cancer Genomics (https://www.cbioportal.org/study/summary?cancer_study_id=prad_tcga_pub). Transcriptomes and corresponding clinical information from Memorial Sloan-Kettering Cancer Center (MSKCC) were downloaded from cBioPortal (https://www.cbioportal.org/study/summary?id=prad_mskcc) and NCBI GEO under accession GSE21032. Microarray data of LTL331R NEPC tumor model were downloaded from NCBI GEO under accession number GSE59986. Gene expressions in 22RV1, LNCaP95, MSKCC-EF1, and H660 were obtained using GEO accession number GSE154576 (https://www.ncbi.nlm.nih.gov/geo/query/acc.cgi?acc=GSE154575).The RNA sequence data in the present study have been deposited to NCBI's Gene Expression Omnibus (GEO) using the accession number GSE64143. All data are available from the authors upon request. All related figures and tables relating to supplementary information are provided in the Supplementary information file and Supplementary Data 1 excel file. All source data supporting this study and uncropped blot images in this manuscript are provided as Supplementary Data 2.

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

## Acknowledgements

This work was supported in part by grants CA253605 (A.C.G.), CA 225836 (A.C.G.), CA250082 (A.C.G.), DOD PC180180 (A.C.G.), and the U.S. Department of Veterans Affairs, Office of Research & Development BL&D grant number I01BX004036 (A.C.G.), BLR&D Research Career Scientist Award IK6BX005222 (A.C.G.). This work was also supported in part by the UC Davis Comprehensive Cancer Center. A.C.G. is also a Senior Research Career Scientist at VA Northern California Health Care System, Mather, California.

## Author contributions

S.N. and A.C.G. conceived the project and designed the experiments. S.N., J.Z., and A.C.G. developed the methodology. S.N. performed the experiments and acquired the data. S.N. and J.G.Z. performed the bioinformatics analysis. S.N., J.G.Z., A.P.L., L.S.D., A.R.L., and M.S. provided technical and methodology support. W.L., C.F.L., J.C.Y., C.P.E., E.C., H.W.C., A.M.Y., and P.M.G. coordinated the tumor specimen acquisition and provided technical and material support. S.N. and A.C.G. wrote and edited the manuscript. A.C.G. acquired funding and supervised the study.

## Competing interests

The authors declare no competing interests.
