## [Peer Review File · Communications Medicine]

Reviewers' comments:

Reviewer #1 (Remarks to the Author):

The manuscript by Ning et al describes a novel signature that positively correlates with neuroendocrine markers and negatively correlates with PSA in prostate cancer. They provided data showing that this signature (genes) stratify patients with neuroendocrine disease from adenocarcinoma of the prostate. This signature was associated with Gleason score and metastasis and postulated that this signature is an early molecular alteration toward neuroendocrine differentiation in advanced prostate cancer patients after extensive AR-targeted therapies. There is an urgent need to identify patients that can progress to the lethal neuroendocrine prostate cancer post hormone therapy. However, it is well established that NE patients have a transcriptome that it is enriched with lineage plasticity including embryonic stem cell pathways and neuronal differentiation.

As AR-high metastatic CRPC progresses toward AR low, ampicrine and eventually neuroendocrine stage, this sentence is not correct. To date, it still unclear if the ampicrine state is a preamble for neuroendocrine.

The authors utilized C42 MDVR cells that express both AR and AR variant as well as NE markers, are these cells transcriptomically cluster with AR and NE positive patients?

It is not clear from the figure 2C how they got to 239 genes and importantly to the 95 unique genes, they should explain this in great detail since the rest of the paper rely on this finding. Moreover, this should be also explained in methods and figure legend. In figure 2C legend, the authors describe Venn diagram showing the number of the unique and shared upregulated genes between t-SCNC vs. Adenocarcinoma, NEPC vs. CRPC and C4-2B MDVR cells. these upregulated genes are only 7 based on the description. Not all the patients are t-SCNC in these datasets, the authors should correct this. Is this identified neural signature clusters with embryonic stem cell signatures?

In page 4, what do you mean by high grade NEPC. NEPC doesn't have a grading, please explain or omit.

How many genes from the 95 genes signature were shared between cell lines and different data set and how many of them where significantly up regulated? What was the rational for the 3 genes chosen to be further characterized.

The survival data was done in adenocarcinoma patients that not necessary progress to neuroendocrine. The prognostic potential of these 3 genes has nothing to do with the disease studied in this manuscript

It was not clear why the authors singled out ARHGEF2 for further characterization and how does targeting Rho-Rac guanine nucleotide exchange factor 2 (ARHGEF2) affect NE markers. Is this effect is a consequence of cell proliferation?

The authors should be consistent with their nomenclature as well the color coding for the phenotype should be consistent it is confusing.

Reviewer #2 (Remarks to the Author):

The authors describe the methodology to identify and validate a 95 neural lineage signature that illustrates early molecular alterations of aggressive post treatment neuroendocrine disease. The authors have compared this signature with other existing ones (Aggarwal, Beltran).

1. Over all I found the introduction very long. Paragraph 3 seems not fitting properly there.

2. One valuable xenograft model that has been used for t-SCPC development is the LTL331R system. ([https://www.cell.com/cell-reports/comments/S2211-1247\(15\)00752-4](https://www.cell.com/cell-reports/comments/S2211-1247(15)00752-4)). Data is available at GSE59986. It is recommended to test the signature in this model as it may represent a closer biology to neuroendocrine dedifferentiation.

3. If the authors repeat their discovery effort using the LTL331R system, how will the result be different?

4. Another interesting study that characterized NEPC signatures and validated it in multiple other cohorts is by Alshalalfa et al (<https://onlinelibrary.wiley.com/doi/full/10.1002/ijc.32430>). They have developed a 212 gene signature that might be worth comparing it with the 95 gene signature. One interesting analysis conducted by Alshalalfa is that they showed that their signature can separate small cell lung carcinoma (SCLC) from non-SCLC. Maybe the authors can test this signature in other tumor types as well.

5. One important aspect of building such a model is the clinical utility. How can the authors introduce this signature in clinical practice. Can the authors test this in aggressive adenocarcinoma to test if it can pick up adenocarcinoma with early NE differentiation?

6. The discussion seems to cover more about literature rather than discussing the current result. I would rewrite it focusing on how this signature is different than Beltran, Tsai et al, Aggarwal, Alshalalfa et al in terms of biology.

We appreciate all the comments by the reviewers and have incorporated their suggestions in the revised manuscript. The following responses address the reviewers' comments in a point-by-point manner.

Reviewer #1 (Remarks to the Author): Referee #1: Prostate cancer, NEPC

The manuscript by Ning et al describes a novel signature that positively correlates with neuroendocrine markers and negatively correlates with PSA in prostate cancer. They provided data showing that this signature (genes) stratify patients with neuroendocrine disease from adenocarcinoma of the prostate. This signature was associated with Gleason score and metastasis and postulated that this signature is an early molecular alteration toward neuroendocrine differentiation in advanced prostate cancer patients after extensive AR-targeted therapies. There is an urgent need to identify patients that can progress to the lethal neuroendocrine prostate cancer post hormone therapy. However, it is well established that NE patients have a transcriptome that it is enriched with lineage plasticity including embryonic stem cell pathways and neuronal differentiation.

As AR-high metastatic CRPC progresses toward AR low, ampicrine and eventually neuroendocrine stage, this sentence is not correct. To date, it still unclear if the ampicrine state is a preamble for neuroendocrine.

Answer: We appreciate the reviewer's suggestion. In the revised manuscript, we have included the citation ¹ regarding the sentence mentioned and make the corresponding changes as reviewer suggested. It was true that it's still lacking sufficient epidemiological data to support ampicrine state is a preamble for NEPC.

Here we are trying to cite the research from Colm Morrissey and their colleagues that increased proportion of treatment-resistant CRPC metastases revealed AR-null phenotypes including small cell or neuroendocrine or double negative phenotypes, which may indicate a disease continuum for AR inhibiting therapies in prostate cancer. Other studies also reported that metastatic CRPC could demonstrate a loss of classical AR signaling upon the AR targeted therapies^{2,3}, inducing the emergence of the cases of AR-null and AR-low metastases.

The authors utilized C42 MDVR cells that express both AR and AR variant as well as NE markers, are these cells transcriptomically cluster with AR and NE positive patients?

Answer: Yes. According to RNA-sequencing and Western blots (Fig1A), C4-2B MDVR cells demonstrates an increased for NE markers including CHGA, NSE and SYP, as well as full-length AR and its variants. This is consistent with other groups report⁴. As we investigated the gene expression of the NE signature genes according to the Beltran⁵ and Aggarwal⁶, these gene signatures were not fully appreciated in C4-2B MDVR due to the disease stage still at adenocarcinoma. However, a common gene program associated to neuron differentiation and stemness were enriched in both C4-2B MDVR and NEPC patient cohorts, which indicates an early NE lineage in AR inhibitors treated CRPC (C4-2B MDVR). It was true that when we utilized the patient dataset from Labrecque study¹ to investigate our NLS signature genes, AMPC groups with both AR hallmark genes and NE markers, in consistent with our NLS 95 genes. Although it was hard to compare C4-2B MDVR/Parental versus the AR+/NE+ patients, NLS 95 genes were significantly enriched in MDVR cells (NES=2.75, FDR-q<0.001). Besides, we examined more cell lines including CRPC enzalutamide resistant 22RV1, LNCaP 95 vs. NIH-H660 and MSCKK-EF1 cell lines, which demonstrate that C4-2B MDVR, 22RV1 and LNCaP95 cells carry increased neural lineage genes compared in comparison with antiandrogen sensitive C4-2B cells, and could cluster with AR and NE positive patients as described in Fig 4C.

It is not clear from the figure 2C how they got to 239 genes and importantly to the 95 unique genes, they should explain this in great detail since the rest of the paper rely on this finding. Moreover, this should be also explained in methods and figure legend. In figure 2C legend, the authors describe

Venn diagram showing the number of the unique and shared upregulated genes between t-SCNC vs. Adenocarcinoma, NEPC vs. CRPC and C4-2B MDVR cells. these upregulated genes are only 7 based on the description. Not all the patients are t-SCNC in these datasets, the authors should correct this.

Answer: As reviewer suggested, we modified the Fig2C to better specify how we get 95 unique genes as neural lineage signature. Correspondingly, the paragraph and the figure legend. 7 genes were commonly shared among the two patient Beltran/Aggarwal and C4-2B MDVR cells, however we enlarged the pool of genes (57 overlapped genes in two patient datasets, 13 genes shared in Aggarwal vs. MDVR and 18 genes shared in Beltran vs. MDVR), thus to include the neural lineage genes increased on the early occurrence of AR treated prostate adenocarcinoma.

We agree with reviewer that patients from Beltran study are not all t-SCNC. The patients from Beltran study are CRPC patients with NE features, and they proposed a possible model of tumors evolved from prostate adenocarcinoma to CRPC-NE⁵, which was further verified in LNCaP cells that it acquired CRPC-NE features after long-term treatment with enzalutamide. This supports the rationale for us to incorporate their CRPC-NE cohort to screen neural lineage genes in prostate adenocarcinoma stage.

Is this identified neural signature clusters with embryonic stem cell signatures?

Answer: Yes. In our neural lineage pathway enrichment analyses, a total of 22 pathways were screened, and the gene set of Human Embryonic Stem Cell Pluripotency from Pathcards was selected as one of the final eight pathways to identify neural lineage signature. GSEA analyses showed that genes regulating human embryonic stem cell pluripotency was both enriched in Beltran, Aggarwal SC/NEPC patients and C4-2B MDVR cells. The final 95 NLS genes were screen the differentially expressed genes in SC/NEPC patients in this gene set.

In page 4, what do you mean by high grade NEPC. NEPC doesn't have a grading, please explain or omit.

Answer: We appreciated the reviewer's suggestion and have modified the wording. High-grade NEPC was referred to the original publication⁷ which defined to include small cell NEPC and large cell NEPC. We agree with reviewer, out of their context, it would be more accurate to describe as aggressive NEPC instead of high-grade NEPC.

How many genes from the 95 genes signature were shared between cell lines and different data set and how many of them where significantly up regulated? What was the rational for the 3 genes chosen to be further characterized.

Answer: As mentioned in previous reply, the 13 + 18 +7 genes were shared between C4-2B MDVR vs. either of the two patient SC/NEPC data, which were all reaching a significance ($p < 0.05$). 7 genes shared among the patient data and cell line data were of the most interest to further characterize. We tried to interrogate if these 7 hub genes could serve as early detection of neural lineage plasticity in early stage PCa. In TCGA⁸ and MSKCC⁹ the expression of neural lineage signature 95 genes was not significant enriched in TCGA and MSKCC cohort, which is reasonable that the patients are primary prostate cancer instead of terminate neuroendocrine stage after the antiandrogen treatment. However, the analyses revealed certain similarities and some differences between primary stage and metastatic prostate cancer. In TCGA cohort, ARHGEF2, LHX2, EPHB2 and FYN were observed an increase trend along with higher Gleason Score, while GNG4, DPYSL3 and APBB2 did not show obvious upregulation in TCGA. Therefore, we focused on these 3 genes for the subsequent studies.

The survival data was done in adenocarcinoma patients that not necessary progress to neuroendocrine. The prognostic potential of these 3 genes has nothing to do with the disease studied in this manuscript.

Answer: We agree with reviewer that the patients were mostly adenocarcinoma and revised accordingly. In MSCKK study⁹, it includes 19 patients with metastases. Wassim study (SU2C/PCF Dream Team)¹⁰ has 14 patients diagnosed as adenocarcinoma with NE features. These patients' data could serve as cohorts to observe the potential clinical indicator for the neural lineage signature genes.

It was not clear why the authors singled out ARHGEF2 for further characterization and how does targeting Rho-Rac guanine nucleotide exchange factor 2 (ARHGEF2) affect NE markers. Is this effect is a consequence of cell proliferation?

Answer: We appreciated the reviewer's comment. ARHGEF2 has a higher expression level in H660 cells compared with the other genes. In our viability assay by knocking down these three genes, ARHGEF2 seems have most strong effects on inhibiting H660 cell growth.

The authors should be consistent with their nomenclature as well the color coding for the phenotype should be consistent it is confusing.

Answer: We appreciated the reviewer's comment. We have made corresponding changes in the revised manuscript.

Reviewer #2 (Remarks to the Author): Referee #2: Omics, bioinformatics

The authors describe the methodology to identify and validate a 95 neural lineage signature that illustrates early molecular alterations of aggressive post treatment neuroendocrine disease. The authors have compared this signature with other existing ones (Aggarwal, Beltran).

1. Overall I found the introduction very long. Paragraph 3 seems not fitting properly there.

Answer: We appreciated the reviewer's comment. We have made corresponding changes in the revised manuscript.

2. One valuable xenograft model that has been used for t-SCPC development is the LTL331R system. ([https://www.cell.com/cell-reports/comments/S2211-1247\(15\)00752-4](https://www.cell.com/cell-reports/comments/S2211-1247(15)00752-4)). Data is available at GSE59986. It is recommended to test the signature in this model as it may represent a closer biology to neuroendocrine dedifferentiation.

3. If the authors repeat their discovery effort using the LTL331R system, how will the result be different?

Answer: We appreciated the reviewer sharing this dataset. It was strikingly significant to see the enrichment of NLS 95 genes in LTL331R NEPC PDX tumors. We have incorporated the corresponding results in supplementary figure 4 to add this experimental model as another strong evidence to support our neural lineage feature in the progression of NEPC tumors.

4. Another interesting study that characterized NEPC signatures and validate it in multiple other cohorts is by Alshalalfa et al (<https://onlinelibrary.wiley.com/doi/full/10.1002/ijc.32430>). They have developed a 212 gene signature that might worth comparing it with the 95 gene signature. One interesting analysis conducted by Alshalalfa is that they showed that their signature can separate small cell lung carcinoma (SCLC) from nonSCLC. May be the authors can test this signature in other tumor types as well.

Answer: We appreciated reviewer's comment and information. By comparing 212 NEPC signature generated by Alshalafa et al and our NLS 95 genes, a list of 11 genes were found overlapped (PROX1, FYN, CHRNA3, NRCAM, HOXD10, EZH2, LHX2, ARHGEF2, NCAM1, CAMK1D, SYP). Interestingly, 3

genes were the same as our identified 7 hub NLS genes (FYN, LHX2, ARHGEF2), which add more evidence to support our focus on ARHGEF2. It would be interesting to test our NLS 95 genes to other tumor types to see how much similarity in between in the future.

5. One important aspect of building such model is the clinical utility. How can the authors will introduce this signature in clinical practice? Can the authors test this in aggressive adenocarcinoma to test if it can pick up adenocarcinoma with early NE differentiation?

Answer: We appreciated the reviewer's comment. NLS genes were identified based on transcriptomic sequencing data. We are not sure how much feasibility to apply RNA sequencing on patient's tumor samples, which would be valuable not only for a timely diagnosis and also could serve as a prediction for prognosis and treatment response. We are now evaluating the 3 neural lineage genes (ARHGEF2, LHX2 and EHPB) to examine the histochemistry in patient's samples, which could potentially serve as a marker in clinic. This would require more clinical samples to validate if these markers could recognize early NED among the treatment resistant prostate adenocarcinoma.

6. The discussion seems to cover more about literature rather than discussion the current result. I would rewrite it focusing on how this signature is different than Beltran, Tsai et al, Aggarwal, Alshalalfa et al in terms of biology.

Answer: We appreciated the reviewer's suggestions. We have made corresponding changes in the revised manuscript.

References:

1. Labrecque, M.P. *et al.* Molecular profiling stratifies diverse phenotypes of treatment-refractory metastatic castration-resistant prostate cancer. *J Clin Invest* **129**, 4492-4505 (2019).
2. Bluemn, E.G. *et al.* Androgen receptor pathway-independent prostate cancer is sustained through FGF signaling. *Cancer cell* **32**, 474-489. e476 (2017).
3. Tomlins, S.A. *et al.* Integrative molecular concept modeling of prostate cancer progression. *Nat Genet* **39**, 41-51 (2007).
4. Bland, T. *et al.* WLS-Wnt signaling promotes neuroendocrine prostate cancer. *iScience* **24**, 101970 (2021).
5. Beltran, H. *et al.* Divergent clonal evolution of castration-resistant neuroendocrine prostate cancer. *Nat Med* **22**, 298-305 (2016).
6. Aggarwal, R. *et al.* Clinical and Genomic Characterization of Treatment-Emergent Small-Cell Neuroendocrine Prostate Cancer: A Multi-institutional Prospective Study. *Journal of Clinical Oncology* **36**, 2492-2503 (2018).
7. Tsai, H.K. *et al.* Gene expression signatures of neuroendocrine prostate cancer and primary small cell prostatic carcinoma. *BMC Cancer* **17**, 759 (2017).
8. Cancer Genome Atlas Research, N. The Molecular Taxonomy of Primary Prostate Cancer. *Cell* **163**, 1011-1025 (2015).
9. Taylor, B.S. *et al.* Integrative genomic profiling of human prostate cancer. *Cancer cell* **18**, 11-22 (2010).
10. Abida, W. *et al.* Genomic correlates of clinical outcome in advanced prostate cancer. *Proceedings of the National Academy of Sciences* **116**, 11428-11436 (2019).

REVIEWERS' COMMENTS:

Reviewer #1 (Remarks to the Author):

the authors responded appropriately to all comments

Reviewer #2 (Remarks to the Author):

I thank the authors for addressing my comments.